# Ligand-Dependent Downregulation of Guanylyl Cyclase/Natriuretic Peptide Receptor-A: Role of miR-128 and miR-195

**DOI:** 10.3390/ijms232113381

**Published:** 2022-11-02

**Authors:** Madan L. Khurana, Indra Mani, Prerna Kumar, Chandramohan Ramasamy, Kailash N. Pandey

**Affiliations:** Department of Physiology, School of Medicine, Tulane University Health Sciences Center, New Orleans, LA 70112, USA

**Keywords:** GC/NPRA, HEK-293 cells, MA-10 cells, internalization, ANP-NPRA/cGMP signaling

## Abstract

Cardiac hormones act on the regulation of blood pressure (BP) and cardiovascular homeostasis. These hormones include atrial and brain natriuretic peptides (ANP, BNP) and activate natriuretic peptide receptor-A (NPRA), which enhance natriuresis, diuresis, and vasorelaxation. In this study, we established the ANP-dependent homologous downregulation of NPRA using human embryonic kidney-293 (HEK-293) cells expressing recombinant receptor and MA-10 cells harboring native endogenous NPRA. The prolonged pretreatment of cells with ANP caused a time- and dose-dependent decrease in ^125^I-ANP binding, Guanylyl cyclase (GC) activity of receptor, and intracellular accumulation of cGMP leading to downregulation of NPRA. Treatment with ANP (100 nM) for 12 h led to an 80% decrease in ^125^I-ANP binding to its receptor, and BNP decreased it by 62%. Neither 100 nM c-ANF (truncated ANF) nor C-type natriuretic peptide (CNP) had any effect. ANP (100 nM) treatment also decreased GC activity by 68% and intracellular accumulation cGMP levels by 45%, while the NPRA antagonist A71915 (1 µM) almost completely blocked ANP-dependent downregulation of NPRA. Treatment with the protein kinase G (PKG) stimulator 8-(4-chlorophenylthio)-cGMP (CPT-cGMP) (1 µM) caused a significant increase in ^125^I-ANP binding, whereas the PKG inhibitor KT 5823 (1 µM) potentiated the effect of ANP on the downregulation of NPRA. The transfection of miR-128 significantly reduced NPRA protein levels by threefold compared to control cells. These results suggest that ligand-dependent mechanisms play important roles in the downregulation of NPRA in target cells.

## 1. Introduction

Atrial and brain natriuretic peptides (ANP, BNP) belong to the cardiac natriuretic peptide (NP) hormone family, which also consists of C-type natriuretic peptide (CNP), *Dendroaspis* NP (DNP), and urodilatin (URO), structurally related to ANP and BNP but not produced or secreted from the cardiomyocytes. ANP and BNP exert diuretic, natriuretic, and vasorelaxant activities, largely directed towards the reduction of high blood pressure (BP) and cardiovascular homeostasis [1,2,3]. CNP is synthesized in the brain and endothelial cells [4], DNP is present in snake venom; however, traces of it have been found in human blood and tissue [5,6]. URO is mainly produced in the kidney and secreted in urine [7,8,9]. The NPs family is composed of highly similar amino acid sequences containing a 17-member disulfide ring structure, but they are genetically distinct and encoded by separate genes that vary remarkably in genomic sequence [10,11]. ANP and BNP are largely synthesized in the granules of the heart atrium and ventricles, circulate in the plasma, and display the most variability in the primary structure, whereas CNP is highly conserved [2,10]. Both ANP and BNP elicit vascular, renal, and endocrine effects, resulting in the maintenance of BP, extracellular fluid volume, and cardiovascular homeostasis [3,12,13]. Three subtypes of natriuretic peptide receptors (NPR) have been recognized: -A, -B and -C (NPRA, NPRB, and NPRC) [2,3,14]. NPRA and NPRB consist of an extracellular ligand-binding domain, a single transmembrane region, an intracellular protein kinase-like homology domain (protein-KHD), and a guanylyl cyclase (GC) catalytic domain [15,16,17,18]. NPRA and NPRB are also referred to as GC-A and GC-B receptors, respectively, and exhibit a high degree of similarity in their primary sequence structures [16,17,19]. GC-A/NPRA is primarily considered the biological receptor for ANP and BNP, since most of the physiological effects of these peptide hormones are elicited by binding to this receptor and generating the second messenger cGMP [3,20,21,22,23]. CNP binds to NPRB, which also generates cGMP. All three NPs are thought to generally bind to NPRC, which lacks intracellular protein-KHD and GC activity but exhibits certain biological and pharmacological responses [19,24,25,26,27].

The downregulation of the membrane receptors is an important phenomenon, which controls the level of receptor function, however, the mechanisms of downregulation of NPRA are not well established. The activity and expression of NPRA is assessed and evaluated mainly through ANP-dependent GC activity and cGMP accumulation [28,29,30,31]. Earlier studies have reported that NPRA and GC activity are regulated by a number of factors, including the ligands (ANP and BNP) [24,29,32,33], growth factors such as transforming growth factor-β1 (TGF-β1), certain neurotransmitter action such as beta-2 adrenergic stimulation, phorbol ester [34,35,36,37], and physiological and pathophysiological milieu [38,39,40]. Physiological changes in Na^+^ and fluid volume in intact animals are known to raise circulating levels of ANP, which has been suggested to decrease the binding of ^125^I-ANP to NPRA [41,42]. The downregulation of NPRA has been predicted in a number of cell systems, including Leydig tumor (MA-10) cells [28,29], recombinant human embryonic kidney-293 (HEK-293) cells [29], and vascular smooth muscle cells (VSMCs) [24,43,44,45], but the underlying mechanisms have not been determined.

Evidence suggests that microRNAs (miRs) have emerged as the essential posttranscriptional regulators of gene expression and receptor activity [46,47,48,49]. They seem to function as sensors of mechanical and cellular changes, linking dynamic physiological processes with gene regulation [50,51]. In general, miRs seem to be involved in various biological processes, and their dysregulation often results in impaired cellular function and disease condition. Specifically, the deletion of miR-128 has been shown to promote proliferation of postnatal cardiomyocytes and cell cycle reentry of adult cardiomyocytes, thereby reducing the fibrosis and cardiac dysfunction in disease conditions [52]. On the other hand, miR-425 has been suggested to repress the production of ANP in cardiac myocytes [46,49]. On the other hand, both miR-425 and miR-155 seem to cooperatively exhibit positive stimulatory role in the ANP expression and cGMP production [53]. Similar to miR-128, a pro-apoptotic role of miR-195 has been described in cardiomyocytes, and Sirt1 was identified as a direct target of miR-195 [54]. Both miR-128 and miR-195 show promise as targets for cardiovascular disease prognosis, prevention, diagnosis, and therapy. In the present study, we selected miRs to determine their roles in the biological action of ANP/NPRA. Specifically, we selected miR-128 and miR-195, which have been used to delineate the role in the downregulation of NPRA. Further, we examined the role of NPs, protein kinase G (PKG), and miRs in the ANP-dependent downregulation of NPRA, GC activity, and cGMP accumulation, utilizing recombinant and/or native endogenous receptor proteins.

## 2. Results

### 2.1. Effect of ANP, BNP, and CNP Treatment on the Downregulation of NPRA

ANP pretreatment led to a significant decrease in ^125^I-ANP binding and downregulation of NPRA in HEK-293 cells overexpressing a high density of recombinant NPRA, and in MA-10 cells expressing high density of endogenous NPRA. Up to 24 h of pretreatment with 100 nM ANP caused a time-dependent reduction in ^125^I-ANP binding in HEK-293 cells (90%) (Figure 1A). Pretreatment of MA-10 cells with 100 nM ANP up to 24 h also caused an 80% reduction in ^125^I-ANP binding to NPRA (Figure 1B). We visualized the internalization of eGFP-tagged receptor after cells were treated with 100 nM ANP for 5, 10 and 30 min (Figure 1C). Cells treated with 100 nM ANP showed the appearance of endocytic vesicles after 5 min; these vesicles gradually decreased over 30 min, which is a visual characteristic of internalized membrane receptors within the cytoplasm. In contrast, endocytic vesicles were absent in untreated control cells. At least 100 untreated or treated cells (three coverslips per experimental condition) were scored using a confocal microscope (Leica) with a 63 × 1.4 NA Plan-Apochromat oil immersion objective lens visualizing downregulation of receptor protein (Figure 1C). The magnified images of areas have been indicated in red squares and highlighted changes in the images have been quantified (Figure 1C,D). Pretreatment with 100 nM BNP also reduced ^125^I-ANP binding to HEK-293 and MA-10 cells (Figure 2A,B). On the other hand, pretreatment with 100 nM of either CNP or c-ANF (a truncated form of ANP), which bind to NPRB and NPRC, respectively, had no effect on the ^125^I-ANP binding in HEK-293 or MA-10 cells (Figure 2A,B). The pattern of ANP downregulation was also dependent on cell type. In the recombinant HEK-293 cells, the decrease in binding was faster compared to MA-10 cells expressing endogenous receptors.

### 2.2. Comparative Analysis of the Effect of ANP, NPRA Antagonist A71915, and PKG Inhibitor, KT-5823 on the Downregulation of NPRA

To determine whether ANP has a direct effect on GC activity of its receptor, we quantified GC activity following ANP pretreatment in HEK-293 and MA-10 cells (Figure 3). We observed a time- and dose-dependent decrease in GC activity in the membrane preparation from HEK-293 and MA-10 cells (Figure 3A,B). Pretreatment with 1 μM A71915 (GC inhibitor) blocked the downregulation of NPRA as assessed by ^125^I-ANP binding in both HEK-293 and MA-10 cells (Figure 4A). We further examined the involvement of PKG in the regulation of NPRA by ANP. Pretreatment with KT-5823 (PKG inhibitor) for 24 h enhanced the inhibitory effect of ANP on the ^125^I-ANP binding in both HEK-293 as well as in MA-10 cells (Figure 4B). We determined the effect of ANP-pretreatment on the intracellular accumulation of cGMP in these cells. ANP time-dependently stimulated the intracellular accumulation of cGMP at 0.5, 1, 2, 4, and 8 h, with maximum levels occurring at 30 min before declining until 8 h (Figure 5A,B). 

### 2.3. Effect of ANP Pretreatment on the Synthesis and Immunofluorescence Analysis of Second Messenger cGMP

We determined the effect of ANP to estimate the visual production of cGMP, we performed immunofluorescence of cGMP, which showed that the intracellular accumulation of cGMP occurred concurrently during receptor internalization and trafficking. Intracellular accumulation of cGMP was visualized using fluorescence intensity by measuring the cGMP immunofluorescence at different time points in intact cells that were treated with 100 nM ANP at 1, 5, 10, 15, and 30 min (Figure 5C). Increased staining was observed in the treated cells, along with significant enhancement of cGMP immunofluorescence intensity at 1 min (two-fold), 5 min (30-fold), 10 min (45-fold), 15 min (30-fold), and 30 min (25-fold) (Figure 5D). As noted above, immunofluorescence intensity showed a diffused and decreasing trend after 10 min.

### 2.4. Effect of PKG Activator, CPT-cGMP and Phosphorylation Inhibitor, Okadaic Acid on ANP-Binding and GC Activity

Pretreatment with 8-(4-chlorophenylthio)-cGMP (CPT-cGMP) (PKG activator) enhanced the ^125^I-ANP binding as well as GC activity of the receptor in treated MA-10 cells (Figure 6A,B). Okadaic acid (OA) blocked the enhancement of ^125^I-ANP binding in MA-10 cells following CPT-cGMP (1 μM) treatment for 30 min (Figure 6C). These results indicate that the pretreatment of cells with ANP caused the downregulation of NPRA, which was accompanied by a decrease in ^125^I-ANP binding, GC activity, intracellular accumulation of second messenger cGMP, and cGMP-dependent PKG activity.

### 2.5. Effect of miR-128 and miR-195 on the NPRA Protein Levels

To confirm the role of miR-128 and miR-195 in *Npr1* regulation, the recombinant HEK-293 cells were transfected with the corresponding miR precursor sequences cloned into the pCMV-MIR-GFP vector and treated with ANP for varying time periods (short-term and long-term). After short-term (0–30 min) treatments with ANP, Western blot and densitometry analysis showed that the expression of NPRA was significantly decreased after transfection with the control miR (Figure 7A,B). However, decreased expression of NPRA was noted after transfection with the miR-128 (Figure 7C,D) and miR-195 (Figure 7E,F). Interestingly, miR-128 significantly diminished the expression of NPRA by almost 70% compared to control miR-transfected cells. Likewise, miR-195 also significantly reduced the expression of NPRA by almost 55% compared to control cells. To further determine the effect of ANP-induced downregulation of NPRA, we treated cells with prolonged (0–24 h) exposure to ANP. Western blot and densitometry analysis were used to show the expression of NPRA after transfection with the control miR (Figure 7G,H). A significant reduction was observed in the expression of NPRA after transfection with miR-128 (Figure 7I,J) and miR-195 (Figure 7K,L; 78% and 40%, respectively) compared to control cells. Moreover, ANP treatment markedly inhibited NPRA protein levels in all transfected cells within 1 h of treatment, and maximum inhibition was observed at 8–24 h.

## 3. Discussion

The present findings demonstrate that pretreatment of HEK-293 cells (expressing recombinant NPRA) and MA-10 cells (expressing endogenous NPRA) with ANP caused dose- and time-dependent downregulation of this receptor protein. Our results show that the downregulation of NPRA was accompanied by decreased receptor binding, reduced GC activity, and a subsequent decrease in intracellular accumulation of cGMP. The inhibition of GC activity blocked the effect of ANP on NPRA, while the inhibition of PKG activity enhanced ANP’s effect on the downregulation of NPRA; however, the stimulatory effect of PKG blocked the ligand-dependent downregulation of this receptor protein. In addition to its physiological effects on natriuresis, diuresis, and vasodilation, ANP, promotes the internalization and turnover of its cell surface receptor GC-A/NPRA [28,29,31]. Homologous downregulation drives the loss of ligand binding capacity on the plasma membrane following prolonged exposure of the cell to the ligand [24,28]. The present results indicate that the net loss of receptors on the cell surface seems to follow the ligand-mediated receptor internalization. We assessed the downregulation of GC-A/NPRA by ANP binding, GC activity, intracellular accumulation of cGMP formation, and/or the internalization of ligand-receptor complexes inside the cell. Both HEK-293 and MA-10 cells contained predominantly NPRA, and its density greatly decreased after a longer period of pretreatment with ANP.

Based on our current findings, we proposed that ANP-mediated downregulation of NPRA seem to involve two different mechanisms: (i) the removal of receptor from the plasma membrane via internalization and degradation of receptor proteins in the lysosomes; and (ii) a reduced level of NPRA due to decreased mRNA synthesis of the receptor at the transcriptional level. Downregulation of NPRA seems to require several sequential steps: the binding of ANP to NPRA, removal of receptor from the plasma membrane, internalization inside the cell, and degradation into the intracellular compartment, likely in the lysosomes. Treatment of cells with A71915 (GC inhibitor) blocked the downregulation of NPRA, probably by inhibiting the internalization of this receptor protein. Our previous studies have provided evidence for the role of short-sequence motifs GDAY and FQQI in the C-terminal domain of NPRA in ligand-mediated internalization of this receptor protein [55,56,57,58]. The ligand-mediated downregulation of GDAY/AAAA and FQQI/AAAA mutant receptors decreased by 48% and 56%, respectively, compared to control wild-type NPRA. In the present study, we observed that the pretreatment of NPRA with ANP caused a time- and dose-dependent decrease in ANP binding to the receptor. The decrease in receptor protein levels, as estimated by Western blotting, further confirmed the downregulation of GC-A/NPRA in response to prolonged treatment with ANP. The results also showed a decrease in GC activity of NPRA after pretreatment with ANP in both HEK-293 and MA-10 cells. Pretreatment of these cells with A71915 (NPRA antagonist) blocked ANP’s effect on downregulation of NPRA. These results suggest that A71915 binds to NPRA, antagonizing the effect of ANP by preventing its binding to receptors. A71915 was unable to mimic the effect of ANP on either GC activity or intracellular generation of cGMP and was unable to downregulate the receptor. Previously, we reported that the receptors with the mutation in GDAY and FQQI motif in the C-terminal domain of NPRA inhibited its internalization, GC activity of receptor, and intracellular accumulation of cGMP [3,56]. Those findings further support our current results, suggesting a role of C-terminal domain in the downregulation of NPRA [55,56,59].

In the current study, the inhibition of ^125^I-ANP binding in the presence of a PKG inhibitor indicates a role of PKG in downregulation of NPRA. Our results suggest that ANP-dependent downregulation of NPRA decreases PKG activity. Furthermore, the reduction in receptor activity was accompanied by a dose- and time-dependent decrease of NPRA mRNA levels. Our current proposed mechanism of downregulation of NPRA follows the route of decreased ANP binding to NPRA, decreased receptor protein levels, reduced GC activity decreased generation of intracellular cGMP, inhibited PKG activation and reduced phosphorylation of NPRA. We observed that okadaic acid (OA), which inhibits phosphorylation, also blocked the binding of ^125^I-ANP to NPRA, providing further evidence of a role for PKG in ANP binding to the receptor and suggesting that a decrease in PKG activity might lead to downregulation of NPRA. Activation of GC activity of NPRA and generation of cGMP after the ligand binds to the receptor may result in the dephosphorylation of NPRA, endocytosis, and degradation of the ligand-receptor complexes. Prolonged exposure to ANP leads to diminished GC activity of the receptor and decreased generation of cGMP, which might have resulted in the diminished PKG activation. Robust activation of PKG may cause the phosphorylation of NPRA, thereby, blocking the downregulation of the receptor. Earlier reports have suggested that NPRA exists as a phosphoprotein, and that ANP causes time-dependent dephosphorylation of receptor [60,61]; however, these authors later reported that NPRA is phosphorylated in response to ANP exposure [62]. The results of the current study further demonstrate that treating cells with KT-5823 (PKG inhibitor) enhanced the downregulation of NPRA by decreasing ligand binding after KT-5823 treatment. PKG has been shown to mediate the stimulation of L-type calcium current by cGMP [63,64].

The present results suggest that the downregulation of NPRA is associated with multiple mechanisms, including increased internalization and degradation of the ligand-receptor complexes in the intracellular compartments via receptor-mediated endocytotic pathways, miRs, and cGMP/PKG pathway. Intracellular cGMP accumulation initiates the activation of PKG, which phosphorylates different substrate proteins [65]. PKG is the major mediator of cGMP-induced functions of the ANP/NPRA signaling cascade in the adrenal glands, kidneys, heart, and vasculature in response to ligand activation [63,66,67]. Downstream effects of ANP/NPRA/cGMP-induced PKG activation is known to elicit the modulation of L-type calcium channels and cross-talk with heterogeneous receptors, such as G-protein coupled receptors (GPCRs) [68,69,70,71]. PKGs are distributed as membrane-bound, cytosolic, and intranuclear proteins [72,73,74]. Interestingly, it has been demonstrated that PKG is a serine/threonine kinase capable of phosphorylating NPRA in vitro [75,76]. Following ANP treatment, phosphorylation of NPRA seems to be required concurrently to ligand binding, which activates the NPRA, while PKG is recruited to initiate the signaling pathway of the receptor [62,76,77].

Our results suggest that sustained activation of NPRA for longer periods of time leads to protein degradation of the receptor and destabilization of its mRNA. This seems to contribute to a reduction in the density of cell surface receptors on the plasma membrane and eventually leads to the downregulation of NPRA. The downregulation of NPRA might be achieved by the increased concentrations of ANP in the disease conditions, which might create the underlying mechanism in the phenomenon of downregulation of this receptor protein. There may be another phenomenon at work: receptor desensitization, which takes place when the receptor in unable to respond to ANP. Phosphorylation of NPRA at protein-KHD has been reported to be essential for receptor function [62]. On the other hand, prebound PKG might serve as an NPRA kinase, and ANP binding seems to be necessary and sufficient for both the recruitment and the maintenance of membrane-bound PKG [76]. Thus, NPRA-PKG interaction may play a role in the ANP action regulating NPRA. Our results support the notion that phosphorylation of the receptor by the serine-threonine kinase PKG enhances the activity of NPRA by inhibiting its downregulation. We speculate that the manipulation of the NPRA-PKG interaction may favor enhanced ANP binding and signaling in disease states; however, further studies are required to show the significance of PKG in the regulation of NPRA. We speculate that in heart failure, high levels of cardiac ANP and BNP downregulate NPRA, limiting the therapeutic utility of high doses of ANP-based therapies. Our findings indicate that ANP causes the downregulation of the receptor, which is accompanied by a decrease in ANP binding, GC activity, and intracellular formation of cGMP.

The present findings show that miR-128 significantly repressed NPRA protein levels. The miR target predictor programs indicated that miR-128 contains putative recognition sites on the *Npr1* 3′-UTR, which is highly conserved in human and rodent genomes, suggesting that it may regulate similar cellular functions in both species [78,79]. We have shown that NPRA miR also decreased the expression levels of *Npr1* (encoding GC-A/NPRA) by more than 90% [80]. Remarkably, it reduced GC activity and decreased intracellular accumulation of cGMP by 90–95% compared to controls. The proposed schematic representation demonstrates the role of miR-128 and miR-195 in the downregulation of NPRA (Figure 8). Overexpression of miR-128 has also been observed in liver and lung fibrosis and has been suggested to regulate inflammatory and fibrotic genes [81,82]. Significantly increased levels of miR-128 expression have been observed in hypertension patients and with the progression of cardiac hypertrophy compared with controls [83,84].

Similarly, miR-195 also suppressed protein levels of NPRA; however, no putative recognition sites were found in the *Npr1* 3′-UTR, suggesting its indirect role in the regulation of NPRA. miR-195-5p is upregulated in cardiac hypertrophy, induced by angiotensin II (Ang II), but the suppression of miR-195-5p prevented the hypertrophy of H9c2 cardiomyocytes after treatment with Ang II [85,86]. Several miRNAs have been shown to downregulate various membrane receptors, including β_2_-adrenergic receptor (β_2_AR) expression, by translational repression [47]. miR-449b, miR-500, miR-328, and miR-320 have all been shown to downregulate neurokinin receptor (NKR1) at the mRNA and/or protein levels [87]. miR-129-2-mediated downregulation of progesterone receptor has been shown in breast cancer cells in response to progesterone [88]. miR-339 downregulated μ-opioid receptor (MOR) at the post-transcriptional level in response to opioid treatment [89]. Similarly, in rat spinal cords, miR-181a downregulated GABA (Aα-1) receptor subunit at the mRNA and protein levels [48]. miR-29b and miR-181a have been found to downregulate the expression of the norepinephrine transporter (NET) and glucocorticoid receptors (GRs), respectively, in PC12 cells [90]. An understanding of the molecular mechanisms of miR involved in the downregulation of renal and cardiovascular receptors may provide prognostic, diagnostic, and therapeutic strategies in renal and cardiovascular pathophysiology and disease conditions.

In summary, our results demonstrate that prolonged ANP treatment of cells containing either recombinant or native endogenous NPRA caused a time- and dose-dependent decrease in ^125^I-ANP binding, GC activity, and intracellular accumulation of cGMP, while treatment with c-ANF and CNP had no effect. A PKG stimulator caused a significant increase in ^125^I-ANP binding activity, whereas a PKG inhibitor potentiated the effect of ANP by 60% on the downregulation of NPRA. Interestingly, a significant reduction in protein levels of NPRA was observed by miR-128 than miR-195 compared to controls. However, it seems that miR-195 might have an indirect role in the regulation of NPRA. The present findings suggest that ligand-dependent mechanisms involving miRs could play an important role in the regulation of GC-coupled receptor NPRA, which may have important implications in the health and disease.

## 4. Materials and Methods

### 4.1. Materials

ANP (rat 28), BNP, CNP, and cANF (truncated ANF) were purchased from Bachem (Torrance, CA, USA), HEK-293 cell line was obtained from ATCC (Manassas, VA, USA), and recombinant HEK-293 cells overexpressing NPRA were generated in our laboratory [29]. MA-10 cell line was a gift from our early stage collaborator Dr. Mario Ascoli at Vanderbilt University [91]. The transfection reagent LT1 was obtained from Panvera (Madison, WI, USA) and Na ^125^I was purchased from Amersham Biosciences (Piscataway, NJ, USA). Lipofactamine™ and tissue culture supplies were purchased from Invitrogen/Life Technologies (Grand Island, NY, USA), and 3-isobutyl-1 methylxanthin (IBMX) was obtained from Sigma Chemical Co. (St. Louis, MO, USA). DyLightTM405 anti-rabbit IgG (H + L) antibody was obtained from Jackson ImmunoResearch Laboratories. A cGMP assay kit was purchased from Assay Design (Ann Arbor, MI, USA). PD-10 columns were purchased from Pharmacia, Piscataway, NJ. All other chemicals were of reagent grade and were purchased from Sigma Chemical Co (St Louis, MO, USA).

### 4.2. Cell Culture

Cloned mouse MA-10 cell line and HEK-293 cell line were cultured in 75 cm^2^ flasks containing 20 mL of modified Waymouth MB 752/1 medium containing 15% horse serum and Dublecco’s modified Eagle’s medium (DMEM) containing 10% calf serum, respectively, as previously described [28,29]. Large-scale cultures were maintained at 37 °C in an atmosphere of 5% CO_2_ and 95% air_._

### 4.3. Iodination of ANP

ANP was iodinated using the chloramine T method [92] as previously described, with modifications [93]. In brief, 5 μg ANP was incubated in 20 μL sodium phosphate buffer (0.25 M, pH 7.4) for 45 s at room temperature (RT) in the presence of 500 μCi Na^125^I, and 10 µL chloramine T (5 mg/mL). Reaction was stopped by the addition of 20 μL sodium metabisulphite (5 mg/mL). The reaction mixture was applied to PD-10 column, and ^125^I-ANP was purified by eluting with sodium phosphate buffer (0.05 M, pH 7.4) containing 0.25% bovine serum albumin (BSA). The specific activity of ^125^I-ANP ranged from 600–900 μCi/μg.

### 4.4. Receptor Binding Assay

Receptor binding assays were performed as previously described [23,93]. Cells were preincubated with various concentrations of unlabeled ANP for the indicated times, medium was aspirated, and the cells were incubated with 50 mM glycine acetate buffer (pH 3.8) containing 100 mM NaCl for 2 min at RT as previously described [55]. The cells were washed four times with assay medium (containing 0.1% BSA). Binding assays were carried out by incubating 1 × 10^5^ cells with ^125^I-ANP in 2 mL of assay medium. Nonspecific binding was determined in the presence of 100 × excess molar concentrations of unlabeled ANP. The binding assay was carried out at 4 °C for the indicated times. After binding was completed, cells were washed four times with ice-cold assay medium. Cells were lysed in 1 N NaOH at RT, and cell-bound ^125^I radioactivity was counted in a Beckman Gamma 5500 counter (Beckman Instruments, Palo Alto, CA, USA).

### 4.5. Guanylyl Cyclase Assay

Cells were preincubated with various concentrations of ANP for the indicated times. The medium was aspirated and cells were incubated with 50 mM glycine acetate buffer (pH 3.8) containing 100 mM NaCl for 2 min at RT. Cells were washed four times with assay medium containing 0.1% BSA. Plasma membranes were prepared by suspending frozen pellets of cells using five-volumes of sodium phosphate buffer (10 mM, pH 7.4) containing sucrose (250 mM), NaCl (150 mM), EDTA (5 mM), phenylmethylsulfonyl fluoride (PMSF) (1 mM), benzamidine (5 mM), leupeptine (10 μg/mL), and aprotinin (10 μg/mL) as previously described [29,55]. Cells were homogenized in a Dounce homogenizer and centrifuged at 800× *g* for 5 min at 4 °C. Supernatants were collected and pellets resuspended in 5 volumes of buffer, homogenized and centrifuged as described above. Supernatants were pooled and centrifuged at 100,000× *g* for 1 h at 4 °C. Pellets were suspended in HEPES buffer (50 mM, pH 7.4) containing NaCl (150 mM), EDTA (5 mM), PMSF (1 mM), benzamidine (5 mM), leupeptin (10 μg/mL), and aprotinin (10 μg/mL) and recentrifuged at 100,000× *g* for 1 h at 4 °C. The membrane pellet obtained was resuspended in above buffer, and aliquots were frozen in liquid nitrogen and stored at −75 °C until used. GC activity was assayed as previously [70]. For GC assay, 10 uL aliquot of membranes were added to the 100 μL of total reaction mixture containing Tris-HCl buffer (50 mM, pH 7.6), MgCl_2_ (4 mM), GTP (1 mM), BSA (1 mg/mL), creatine phosphate (75 mM), IBMX (2 mM), and creatine phosphokinase (3 units) in the presence of ANP. Reaction mixtures were incubated at 37 °C for 20 min and then stopped with the addition of 900 μL sodium acetate buffer (50 mM, pH 6.2) and by placing the samples in a boiling water bath for 3 min. The amount of cGMP generated was determined by radioimmunoassay [29]. Protein contents were estimated using the Bio-Rad method [94].

### 4.6. Intracellular cGMP Generation

Cells were treated with 100 nM ANP for 0, 0.5, 1, 2, 4 or 8 h in the presence of 0.2 mM IBMX. Cells were washed three times with phosphate-buffered saline (PBS) and scraped in 0.1 M HCl. Cell suspensions were subjected to five freeze–thaw cycles, centrifuged at 10,000× *g* for 15 min at 4 °C, and supernatants were collected. The cGMP was measured using ELISA as previously described [29,31].

### 4.7. Visualization of ANP-Induced Internalization of eGFP-NPRA

The eGFP-NPRA construct was prepared and validated in HEK-293 cell and ANP-induced internalization of eGFP-NPRA was performed as previously described [31,56]. Cells were seeded on cover glass, grown for 2 days in DMEM media containing 10% calf serum and incubated with 100 nM ANP for different time periods. After incubation, cells were washed in PBS, fixed in 4% paraformaldehyde for 30 min, and washed three times with PBS for 5 min each time. Cover glass was allowed to dry, then mounted on glass slide with Vectashield mounting medium (Vector Laboratories, Burlingame, CA, USA) and sealed with Fixogum rubber cement according to the manufacturer’s protocol.

### 4.8. Immunofluorescence of cGMP

To visualize intracellular accumulations of cGMP, immunofluorescence (IF) was performed as previously reported [95,96], with minor modifications [31]. Cells were treated with 100 nM ANP at 1, 5, 10, 15, or 30 min in the presence of 0.2 mM IBMX and then fixed in 4% paraformaldehyde for 30 min, permeabilized in PBS containing 0.1% BSA/0.2% saponin, and incubated for 10 min at RT. Cells were blocked with 1% normal goat serum, 0.1% saponin/1% BSA in PBS for 1 h at RT, then labelled with anti-cGMP antibodies (1:1000) (Antibodies Online Inc., Atlanta, GA, USA, GA Cat number ABIN636408) in blocking buffer overnight at 4 °C. Samples were incubated with secondary antibody goat- anti-rabbit IgG (1:500) conjugated with DyLight™^405^ for 2 h at RT in the dark. Cells were washed three times for 15 min each time in PBS. Cover glass was allowed to dry and mounted on glass slides with Vectashield mounting media (Vector Laboratories), then sealed with Fixogum rubber cement. The immunofluorescence localization of cGMP was done using confocal microscopy as previously reported [31].

### 4.9. Confocal Microscopy

For immunofluorescence analyses, HEK-293 cells were examined and images acquired using a TCS-SP2 confocal laser scanning microscope (Leica Microsystems, Wetzlar, Germany). In all experiments, images of cells (1024 × 1024 pixels) were visualized using the same confocal microscope settings (sequential scans with wavelengths set up as green, 488–510; blue, 400–421) using a 63× Apo-oil immersion objective (NA = 1.4) and 60-μm aperture, using LEICA Scan TCS-SP2 software (Leica Microsystems). For green channel (eGFP-tagged NPRA), excitation was 488 nm and emission was 510 nm, whereas for the blue channel (DyLight™) 405 anti-rabbit antibody for cGMP immunofluorescence, excitation was 400 nm and emission was 421 nm. The pinhole was adjusted to keep the same size of z-optical sections (1-μm *z*-axis) for both channels. In each experiment, cell images were acquired as single mid-cellular optical sections and averaged over eight scans/frame using MetaMorph software as previously described [31].

### 4.10. Predicated Target Sequences of miR and Transfection of Murine (mmu) miR-128 and miR-195 in HEK-293 Cells

Cells were grown at 37 °C in a humidified atmosphere of 5% CO_2_, 95% air and maintained in DMEM with 10% FBS. Cells were seeded in 60-mm dishes at a density producing ~90% confluence. miR expression plasmid miR128-1-pCMV-miR with the mouse precursor sequence of miR-128 (GUUGGAUUCGGGGCCGUAGCACUGUCUGAGAGGUUUACAUUUCUCACAGUGAACCGGUCUCUUUUUCAG) and miR195-pCMV-miR with the mouse precursor sequence of miR-195 (ACACCCAACUCUCCUGGCUCUAGCAGCACAGAAAUAUUGGCAUGGGGAAGUGAGUCUGCCAAUAUUGGCUGUGCUGCUCCAGGCAGGGUGGUGA) were utilized for transfection studies (Creative Biogene, Shirley, NY, USA). The mature sequences are presented in bold and underlined. Using miR target predictor programs (Target Scan release 7.1), we searched for miRs with a predicted target pairing sequence on untranslated region (UTR) of *Npr1* as previously reported [97]. The putative recognition sites (conserved and poorly conserved) of miR-128, the *Npr1* 3′-UTR, and the sequence homology is shown in Table 1. The miR-128 seeding sequence to the *Npr1* 3′-UTR is highly conserved across multiple vertebrate species (mouse, rat, and human). Black vertical lines represent perfect matches in the miR-128 recognition sequence (seeding region) to the 3′-UTR, which is highly conserved across multiple vertebrate species, including mouse (*Mus musculus*), rat (*Rattus norvegicus*), and human (*Homo sapiens*) as indicted by Target Scan (www.TargetScan.org, accessed on 22 June 2021). The bold underlined characters represent the seed sequence for miR-128 (Table 1). In the present studies, we used the mouse *Npr1* sequence [98]. Three micrograms of plasmid DNA were transfected using a Lipofectamine RNAiMAX Reagent transfection kit as per the manufacturer’s protocol (Thermo Fisher Scientific, Waltham, MA, USA). An empty pCMV-miR plasmid was used as a negative control (Creative Biogene, Shirley, NY, USA). Forty-eight hours post transfection, cells were washed with PBS and treated with ANP for short period (0, 2.5, 10, and 30 min) and for long period (0, 1, 4, 8, 12, and 24 h) of incubation. Cells were harvested for Western blot analysis at the indicated time intervals and lysed essentially as described earlier [99].

### 4.11. Preparation of Whole Cell Lysate and Western Blot Analysis

Whole cell lysate was prepared as previously described [70,99]. The protein concentration of the lysate was measured with a Bradford protein detection kit (Bio-Rad, Hercules, CA, USA). Western blot assay was performed as previously reported [23,99]. Cell lysate (50 μg proteins) from each sample was mixed with sample loading buffer and separated using 10% sodium dodecyl sulfate-polyacrylamide gel electrophoresis (SDS-PAGE), then transferred onto a polyvinylidene fluoride membrane as previously described [100]. The membrane was blocked with 1× Tris-buffered saline-Tween 20 (TBST; 25 mM Tris, 500 mM NaCl, and 0.05% Tween 20, pH 7.5) containing 5% fat-free milk for 1 h, then incubated overnight in TBST containing 5% fat-free milk at 4 °C with primary antibody (1:500 dilution). The membrane was treated with corresponding secondary anti-chicken horseradish peroxidase-conjugated antibodies (1:5000 dilutions). Protein bands were developed using Clarity Western ECL Substrate from Bio-Rad and visualized using a FluorChem detection system from ProteinSimple (Santa Clara, CA, USA). The intensity of protein bands was quantified by Alphaview software. Primary antibody for NPRA was produced using our published methods [31,100]. Anti-β-actin antibody (C4) HRP conjugated primary antibody (Santa Cruz Biotechnology, Santa Cruz, CA, USA) was utilized (1:1000 dilution) to visualize β-actin protein as a loading control. 

### 4.12. Statistical Analysis

Statistical analyses were performed using GraphPad Prism software, version 6.0 (San Diego, CA, USA). Results are presented as the mean ± SE. Differences between groups were determined using ANOVA with Dunnett’s multiple comparisons post hoc test. Differences were considered significant at *p* < 0.05.

## Figures and Tables

**Figure 1 ijms-23-13381-f001:**
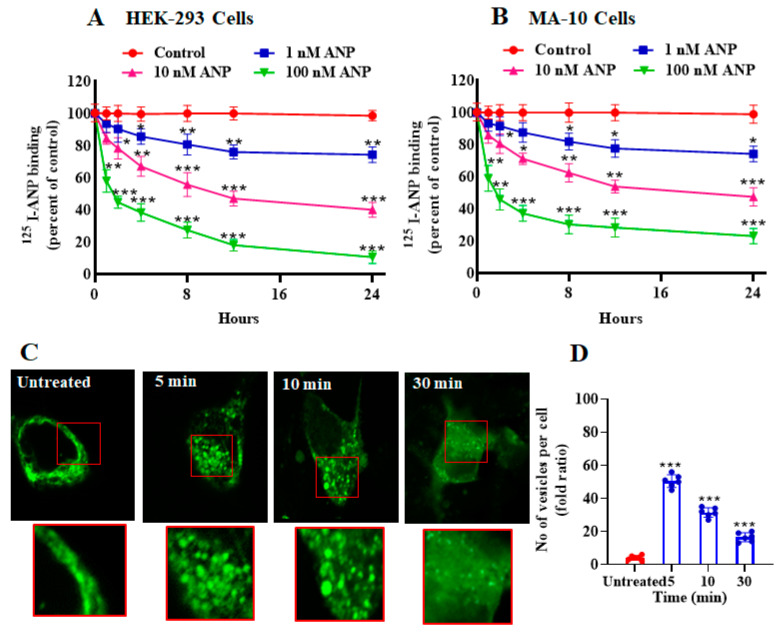
Effect of ANP pretreatment on ^125^I-ANP binding and ANP-induced eGFP-NPRA internalization in stably expressing recombinant HEK-293 cells and endogenously expressing MA-10 cells. (**A**,**B**) Confluent HEK-293 cells and MA-10 cells were pretreated with varying concentrations of ANP for the indicated times at 37 °C. Cells were transferred to 4 °C and ^125^I-ANP binding was done in assay medium containing 0.1% BSA as described under the Section 4 for 30 min at 4 °C. Cells. (**C**) A series of single confocal plane images were taken from HEK-293 cells fixed with 4.0% formaldehyde to visualize the internalization of NPRA after stimulation by 100 nM ANP. The images of mid-focal planes were collected from 5–6 independent experiments. The magnified images of areas have been indicated in red squares. (**D**) The highlighted changes in the images have been quantified and scored. Values are expressed as means ± SE of 6–7 independent experiments. * *p* < 0.05, ** *p* < 0.01, *** *p* < 0.001 (untreated vs. drug-treated group).

**Figure 2 ijms-23-13381-f002:**
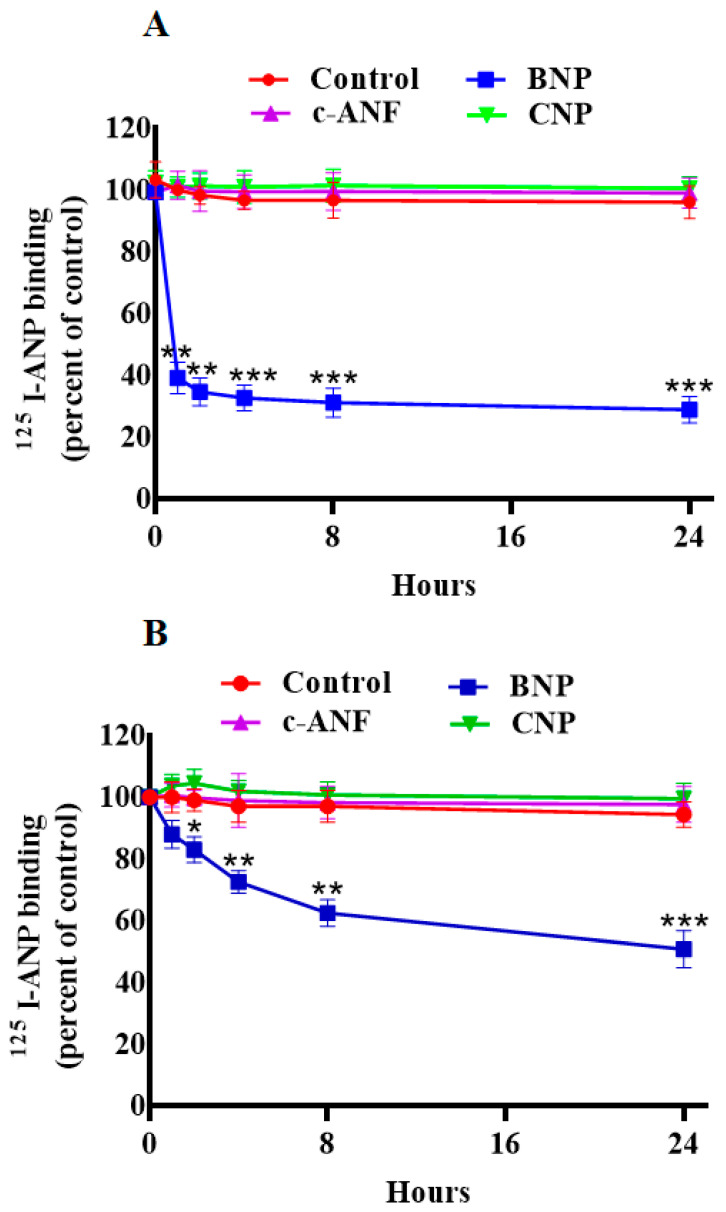
Effect of BNP, c-ANF and CNP pretreatment on ^125^I-ANP binding to NPRA in HEK-293 and MA-10 cells. (**A**) Confluent HEK-293 cells and (**B**) MA-10 cells were pretreated with 100 nM of BNP, c-ANF, or CNP for indicated times at 37 °C. Cells were transferred to 4 °C and receptor binding was carried out as described under the Section 4. Values represents mean ± SE of 7 independent experiments. * *p* < 0.05, ** *p* < 0.01, *** *p* < 0.001 (untreated vs. drug-treated group).

**Figure 3 ijms-23-13381-f003:**
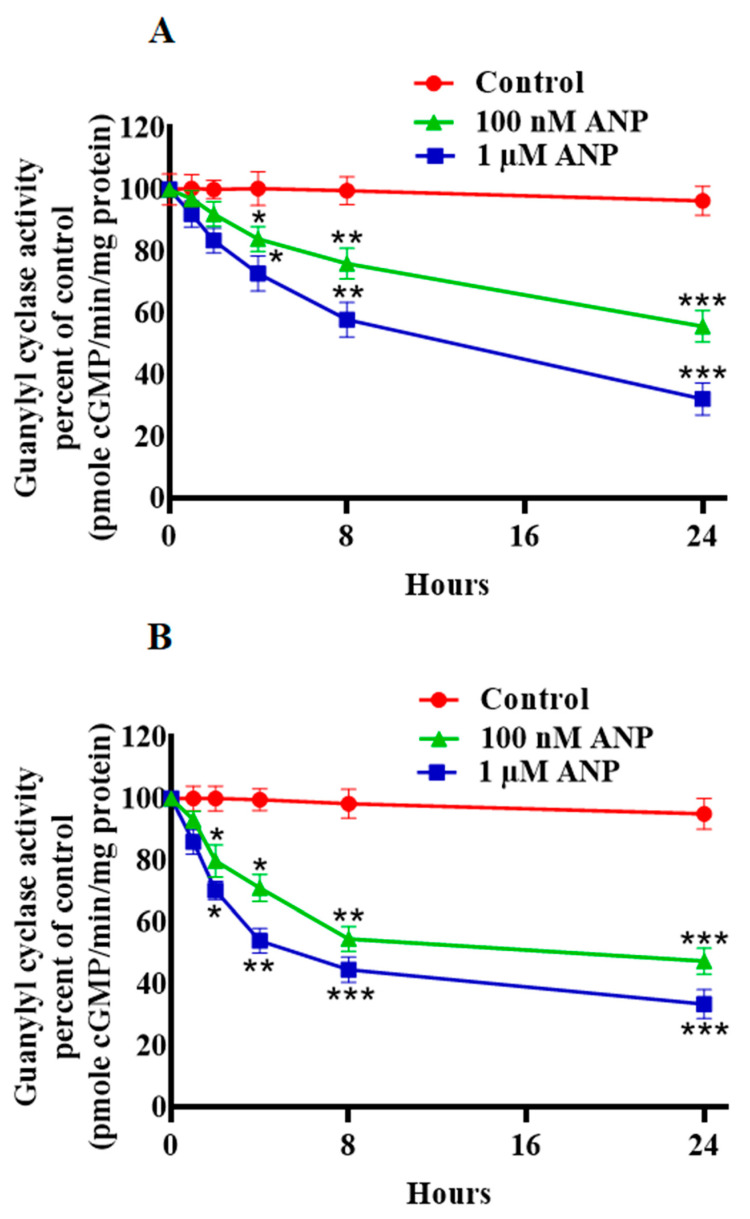
Effect of ANP pretreatment on guanylyl cyclase activity in HEK-293 cells and MA-10 cells. (**A**) HEK-293 cells and (**B**) MA-10 cells were treated with 100 nM ANP for 0, 8, 16, and 24 h. Cells were incubated with glycine acetate buffer (pH 3.8) for 2 min, plasma membranes were prepared. GC activity was determined as described under the Section 4. Values represent mean ± SE of 8 independent experiments. * *p* < 0.05, ** *p* < 0.01, *** *p* < 0.001 (untreated vs. drug-treated group).

**Figure 4 ijms-23-13381-f004:**
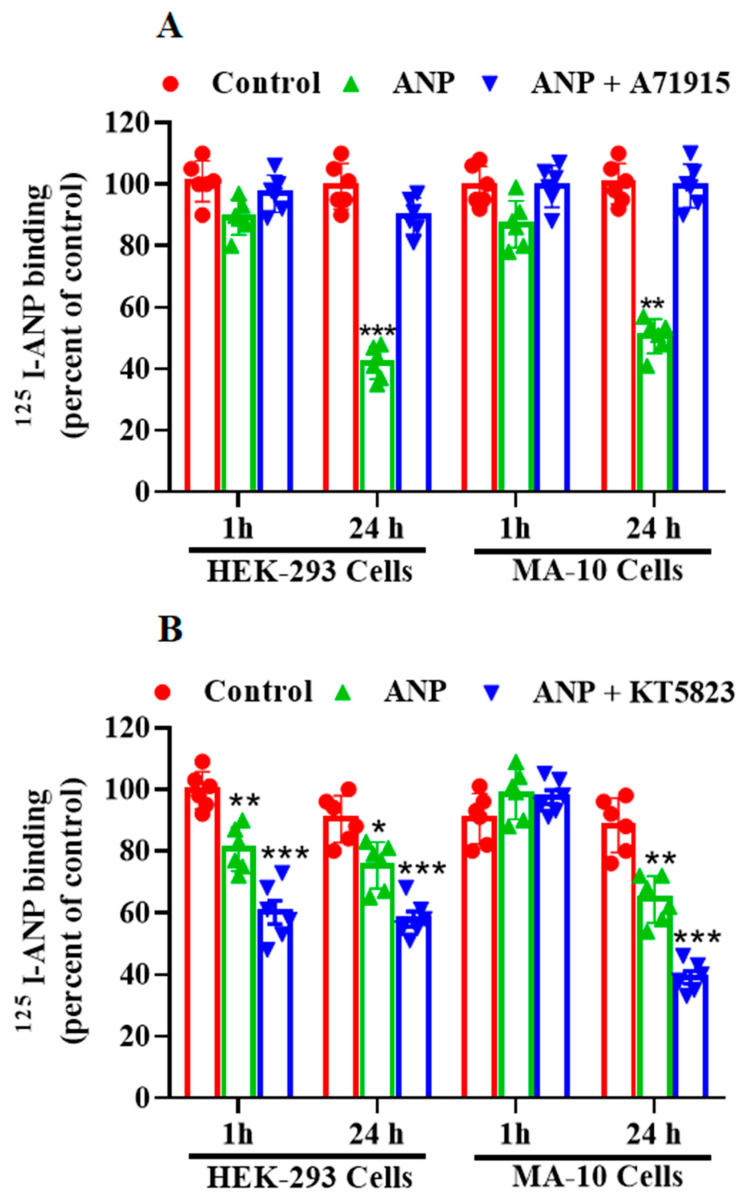
Effects of A71915 and KT 5823 pretreatment on ^125^I-ANP binding to NPRA in HEK-293 and MA-10 cells. Confluent HEK-293 cells and MA-10 cells were pretreated with 100 nM ANP for the indicated times at 37 °C after which cells were transferred to 4 °C. Receptor binding was carried out as indicated under the Section 4. (**A**) Effect of NPRA antagonist, A71915 and (**B**) effect of PKG inhibitor, KT-5823 on ^125^I-ANP binding in HEK-293 and MA-10 cells. The bars represent the mean ± SE of 6 determinations. * *p* < 0.05, ** *p* < 0.01, *** *p* < 0.001 (untreated vs. drug-treated group).

**Figure 5 ijms-23-13381-f005:**
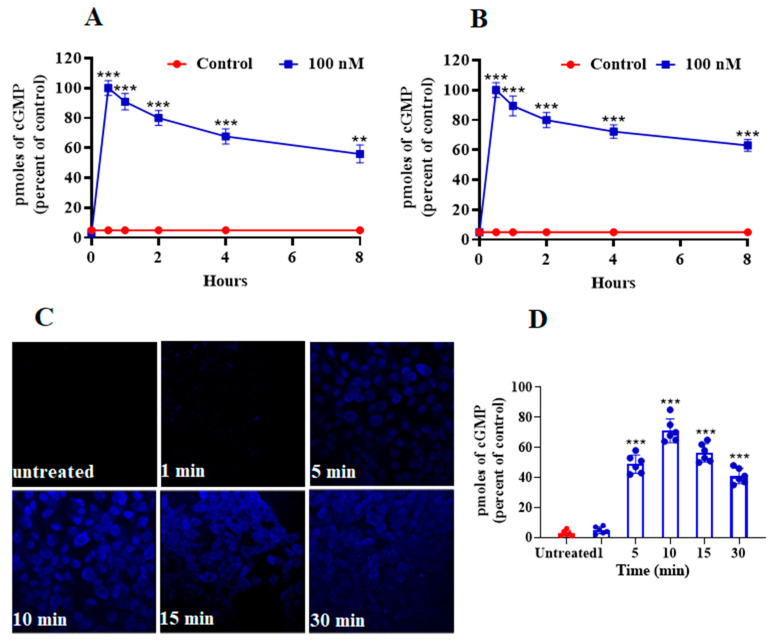
Effect of ANP pretreatment on the synthesis and immunofluorescence analysis of cGMP in HEK-293 and MA-10 cells. To assay the stimulation of intracellular accumulation of cGMP, (**A**) HEK-293 cells and (**B**) MA-10 cells were treated with 100 nM ANP for different times (0, 2, 4, 6 and 8 h) in the presence of IBMX and cGMP was assayed by ELISA as described in the Section 4. (**C**) Untreated cells stained with DyLight™405 anti-rabbit antibody without prior incubation with the first rabbit antiserum. Cells were treated with ANP for 1, 5, 10, 15 or 30 min, which showed the accumulation of cGMP (blue). (**D**) Bars represent the densitometric analysis of the cGMP fluorescence intensities. Values represent means ± SE of 6 independent experiments. Images of mid-focal planes were used from 6 independent experiments. ** *p* < 0.01, *** *p* < 0.001 (untreated vs. drug-treated group).

**Figure 6 ijms-23-13381-f006:**
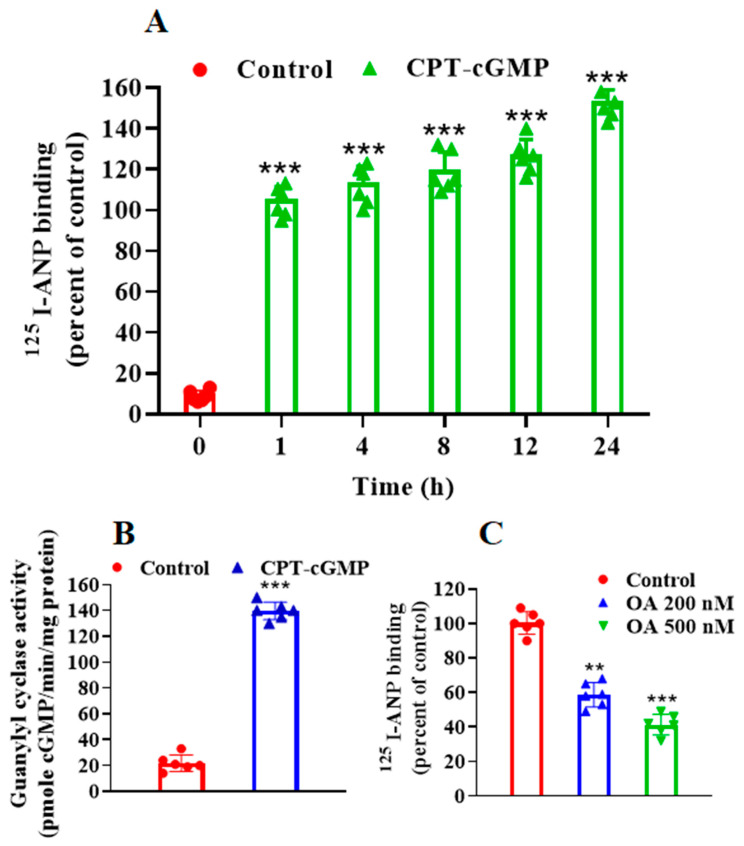
The effect of CPT-cGMP and okadaic acid pretreatments on ^125^I-ANP binding to NPRA and guanylyl cyclase activity. (**A**) Confluent MA-10 cells were pretreated with 1 µM concentrations of 8-(4-chlorophenylthio)-cGMP (CPT-cGMP) for the indicated times at 37 °C and then transferred to 4 °C. Receptor binding was carried out as described under the Section 4. (**B**) MA-10 cells were treated with 1 mM CPT-cGMP for 24 h. The medium was aspirated. Cells were incubated with glycine acetate buffer (pH 3.8) for 2 min and plasma membranes were prepared to assay the GC activity. (**C**) Confluent MA-10 cells were pretreated with CPT-cGMP in the presence or absence of indicated concentrations of okadaic acid (OA) for 24 h at 37 °C. Receptor binding was carried for 30 min at 4 °C as described in the Section 4. Values are expressed as means ± SE of 6 determinations. ** *p* < 0.01, *** *p* < 0.001 (untreated vs. drug-treated group).

**Figure 7 ijms-23-13381-f007:**
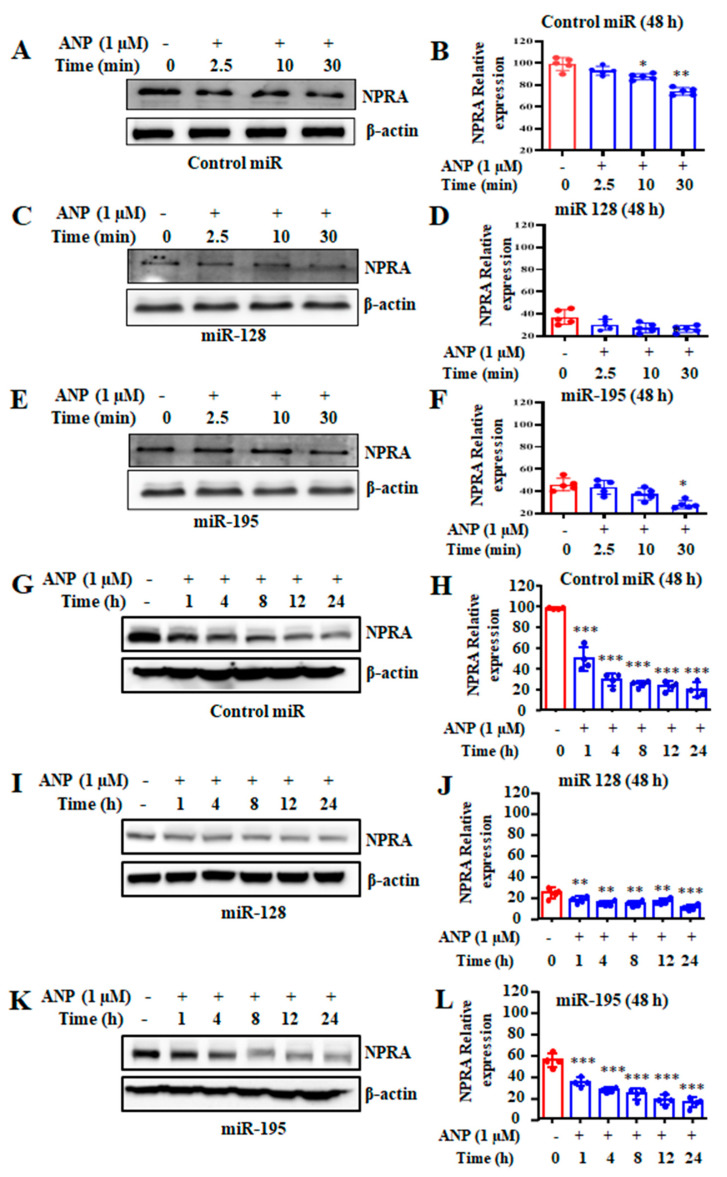
Effect of miRNA overexpression and ANP treatment on NPRA protein levels. HEK-293 cells were transfected with miR-128, miR-195, or vector control (miR control). At 48 h post-transfection, cells were treated with ANP (100 nM) for the indicated times of short-term (0, 2.5, 10, and 30 min) and long-term (0, 1, 4, 8, 12, and 24 h) ANP treatments. After short-term treatments, (**A**) Western blot and (**B**) densitometric analysis of NPRA protein was done with the control miR. β-actin level is shown as loading control, (**C**) Western blot and (**D**) densitometric analysis of NPRA protein levels after transfection with miR-128, and (**E**) Western blot and (**F**) densitometric analysis of NPRA protein levels are shown after transfection with miR-195. Similarly, after long-term treatments, (**G**) Western blot and (**H**) densitometric analysis of NPRA protein levels after transfection with the control miR. (**I**) Western blot and (**J**) densitometric analysis of NPRA protein levels after transfection with miR-128, and (**K**) Western blot and (**L**) densitometric analysis of NPRA protein levels in cells after transfection with miR-195. β-actin level is shown are loading control. Bar represents mean ± S.E. of 5 independent experiments. * *p* < 0.05, ** *p* < 0.01, *** *p* < 0.001 (untreated vs. drug-treated group).

**Figure 8 ijms-23-13381-f008:**
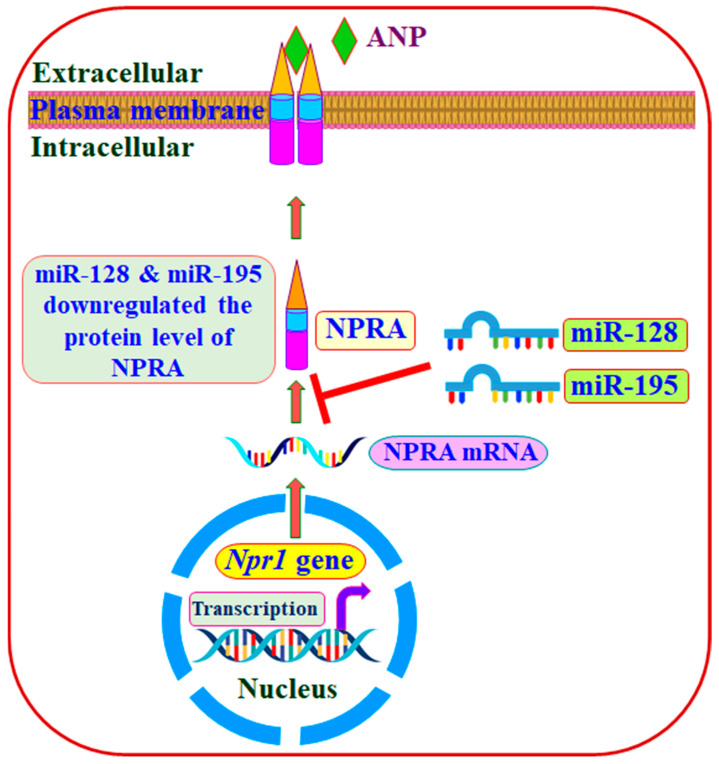
Schematic representation of the role of miR-128 and miR-195 in the downregulation of NPRA. miR-128 significantly downregulated NPRA compared to miR-195. ANP-dependent downregulation of NPRA (ANP-NPRA complex) as compared to control cells, and downregulation of NPRA through miR affects ANP/NPRA/cGMP signaling.

**Table 1 ijms-23-13381-t001:** MiR-128 binding to mouse *Npr1* 3′ UTR and comparison of miR-128 seeding sequence to the *Npr1* 3′-UTR of mouse, rat, and human genes.

**Target Region**	**Predicted Pairing of Target Region (Top)** **and miR (Bottom)**	**Seed Match**
** Pair-Wise Alignment of Conserved Sequences **		
Position of 304–311 of *Npr1* 3′UTR	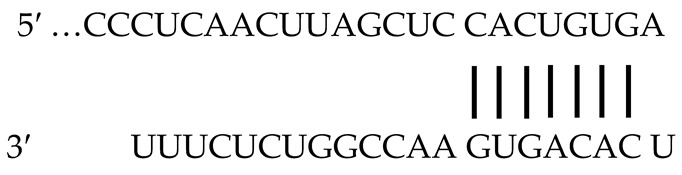	8 mer
mmu-miR-128		
** Pair-Wise Alignment of Poorly Conserved Sequences **		
Position of 91–97 of *Npr1* 3′UTR	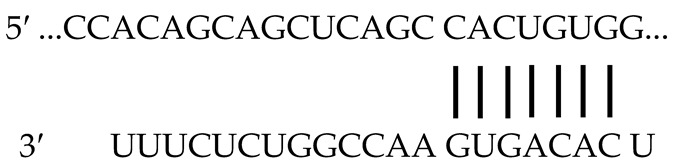	7mer-
mmu-miR-128		
** Alignment of mouse, rat, and human miR-128 recognition sequences **		
M. Musculus	5′-GCCCUCAACUUAGCUC**C-ACUGUGA**CUU-3′	
R. Novergicus	5′-GCCCUCAACUUAGCUC**C-ACUGUGA**CUU-3′	
H. Sapiens	5′-UCCCUCAGCCUUGCUA**C-CCUGUGA**CUU-3′	
miR-128	5′-.........C......aCcuugcu.**C.aCUGuGA**CUU....au.g-3′	

Binding of miR-128 to mouse *Npr1* 3′ UTR. Black vertical lines represent perfect matches of sequence homology between *Npr1* 3′-UTR (top) and miR-128 (bottom). Bold underlined characters represent alignment of the seed sequence for miR-128 and recognition sequence of 3′-UTR of *Npr1* gene in mouse (*Mus musculus*), rat (*Rattus norvegicus*), and human (*Homo sapiens*). UTR, untranslated region; mmu, mouse.

## Data Availability

The corresponding author have all the data available upon request.

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
