# Peer review of "Ligand-Dependent Downregulation of Guanylyl Cyclase/Natriuretic Peptide Receptor-A: Role of miR-128 and miR-195"

_ijms, 2022, doi:10.3390/ijms232113381_

Round 1

Reviewer 1 Report

Dear authors,

The manuscript ‘Ligand-dependent downregulation of guanylyl cyclase/atrial natriuretic peptide receptor-A: role of microRNA-128’ demonstrated an interesting new insight into the cardiac natriuretic peptide hormone family and its role and provide new information about the regulation of miRNA in this topic. Despite the general points are well defended and the results are consistent, there are some important considerations to correct in the manuscript, especially regarding presentation:

INTRODUCTION:

·  Line 35: ‘Dendropsis natriuretic peptide’ should go in lower case.

RESULTS:

·  Results sections must be organised with subsections. The authors present many results in only one section, and this is confusing to follow the manuscript. Regarding organisation, also, it should be considered to reduce the white gap between figures.

·  All figures must show the significant symbols in the graph. Figures 1, 2, 3 and 5A must show significance in the figure in each time point where started to be significant the differences. 

·  The figure legends provide too much methodological information making it redundant, so they must be simplified in all figures. For example, information such as ‘Cells were washed four time with medium containing 0.1% BSA’ or ‘Cells were washed four times with the medium, lysed in 0.5 N NaOH […]’ should be included in the methodology, not in the figure legends.

·  Concerning the colour used for each group, they have been similar in all the figures. For example, if you assigned ‘red’ to Control group, all Control/untreated must be in red (Figure 3, Figure 5D, Figure 8).

·  In Figure 4B, why Control group is not included?

·   Experiments showed in Figure 6 (pretreatment with CPT-cGMP (PKG activator)), and in Figure 8 (miRNAs), why did not done in HEK-293? Also, there is no information provided about the results in Figure 8 coming from MA-10 cells.

·  Figure 6B: Which are the units of the guanylyl cyclase activity? Besides, Figures 6A and 6C have written in the X-axis ‘treatment’, which has no sense to me.

·  Line 214-232 ‘In the present study, we selected microRNAs...NPRA by almost 55% compared to control cells’ provide a mixing between methodology and results. Please, rewrite this paragraph being sure that only results are provided. Also, I consider that Figure 7 and all the information about it, should be in the Methodology sections because ‘it isn’t a result’ obtained from the group, and maybe should be addressed as supplementary information.

·  Some figures are a little bit blurry. I recommend you check the resolution of the figures provided it.

DISCUSSION/CONCLUSION:

·        I recommended adding extra references to support the ideas in:

o   Line 281-284 ‘ANP-mediated downregulation of NPRA […] mRNa synthesis of the receptor at the transcriptional level’.

o   Line 334-336 ‘Vasoconstriction and fluid […] level of plasma ANP’

·        I understand that the reason why you make more emphasis on miR-128 is that this one seems to have a greater effect than miR-195, and miR-195 has an indirect role in the regulation of NPRA. However, I consider the findings about miR-195 part of your, and it should be included as a conclusion.  

MATERIALS AND METHODS.

·  First part of Material and Methods section (Line 402-415) should be included in a specific subsection and also reorganized.

·   In the methodology have been included many references for the same techniques from your laboratory or similar laboratory. Under my concern, the citations should be reduced. For example, Line 452-453 include four references for the same protocol.

·  Check all the sections to avoid typing mistakes. For example, ‘ºC’ (Line 473; 490), ‘CO2’ (Line 506), “MiRNA” (Line 508), etc. Also is described in Line 516 that ‘The nature sequences are present in bold an underline’, however, there is only an underline.

·  In Material and Methods should be included all the extra information provided in the Figure legends and removed from that as I mentioned above.

BIBLIOGRAPHY

·  Please, check all the bibliography and references provided to avoid mistakes. For example, Reference 8 is out of the text, Reference 93 and 94 are the same repeated, etc.

Author Response

RESPONSE TO THE REVIEWER’S COMMENTS

REVIEWER 1:

We thank the reviewer for valuable time to review and provide the valuable comments on our manuscript, which have greatly helped to improve the contents of this manuscript. We also appreciate the reviewer’s comments that the manuscript demonstrates an interesting new insight into the cardiac natriuretic peptide hormone family and its role and provide more information about the regulation of miRNA in this context.

INTRODUCTION:

Comment 1: Line 35: ‘Dendropsis natriuretic peptide’ should go in lower case.

Response: Dendropsis” is the genus, which is written with first letter in upper case and italicized (page 3; line 49).

RESULTS:

Comment 2: Results sections must be organised with subsections. The authors present many results in only one section, and this is confusing to follow the manuscript. Regarding organisation, also, it should be considered to reduce the white gap between figures.

Response: As recommended by the reviewer the “Results” section has been presented with different sections and sub-titles in the revised manuscript. We have rearranged the Figures and panels and have tried to reduce the white gap in the Figures, panels, and labeling of the Figure panels (Figures 1-8) in the revised manuscript.

Comment 3: All figures must show the significant symbols in the graph. Figures 1, 2, 3 and 5A must show significance in the figure in each time point where started to be significant the differences.

Response: As suggested by the reviewer, all Figures now show the significant symbols at different time points and treatments to state the significant differences especially in Figures 1, 2, 3, and 5A.

Comment 4: The figure legends provide too much methodological information making it redundant, so they must be simplified in all figures. For example, information such as ‘Cells were washed four time with medium containing 0.1% BSA’ or ‘Cells were washed four times with the medium, lysed in 0.5 N NaOH […]’ should be included in the methodology, not in the figure legends.

Response: As indicated by the reviewer, the detail methodological information has been deleted and the Figure legends have been revised and made succinct.

Comment 5: Concerning the colour used for each group, they have been similar in all the figures. For example, if you assigned ‘red’ to Control group, all Control/untreated must be in red (Figure 3, Figure 5D, Figure 8).

Response: As recommended by the reviewer, the color for each group of the treatment in the Figures has been assigned with common color for the controls versus treatment groups in the revised manuscript.

Comment 6: In Figure 4B, why Control group is not included?

Response:  As suggested by the reviewer, in Figure 4B, the control group has been included in the revised manuscript.

Comment 7: Experiments showed in Figure 6 (pretreatment with CPT-cGMP (PKG activator)), and in Figure 8 (miRNAs), why did not done in HEK-293? Also, there is no information provided about the results in Figure 8 coming from MA-10 cells.

Response: In these studies, the preliminary results did not show significant differences between MA-10 and HEK-293 cells, thus to avoid some redundancies, we have included only one cell line in these experiments.

Comment 8: Figure 6B: Which are the units of the guanylyl cyclase activity? Besides, Figures 6A and 6C have written in the X-axis ‘treatment’, which has no sense to me.

Response: As indicated by the reviewer, the unit for guanylyl cyclase activity has been provided in Figure 6B. Further, the “Treatments” has been deleted on the X-axis in Figure 6 B and C in the amended manuscript.

Comment 9: Line 214-232 ‘In the present study, we selected microRNAs...NPRA by almost 55% compared to control cells’ provide a mixing between methodology and results. Please, rewrite this paragraph being sure that only results are provided. Also, I consider that Figure 7 and all the information about it, should be in the Methodology sections because ‘it isn’t a result’ obtained from the group, and maybe should be addressed as supplementary information.

Response: As suggested by the reviewer, the sentence (line 214-232, original manuscript) has been corrected and made succinct. The text used for original Figure 7 has been moved to the Materials and Methods (section 4.10). The Figure 7 has been now changed to Table 1 and it has been included in the Methods section with highlighted in bold in the revised manuscript (page 20, 21).

Comment 10: Some figures are a little bit blurry. I recommend you check the resolution of the figures provided it.

Response: We have tried to improve the resolution of the Figures in the revised manuscript.

DISCUSSION/CONCLUSION:

Comment 11: I recommended adding extra references to support the ideas in:

Comment 11 a: Line 281-284 ‘ANP-mediated downregulation of NPRA […] mRNa synthesis of the receptor at the transcriptional level’.

Response: Line 281-284: We have revised the text for the clarity purposes. Based on the current findings, we anticipate that two different mechanisms might be involved in the downregulation of NPRA. We did not find additional references to support our notion, thus we did not add additional references in the revised manuscript (page 9, lines 201-205).

Comment 11 b: Line 334-336 ‘Vasoconstriction and fluid […] level of plasma ANP’

Response: The sentence has been revised to indicate the meaning with more clarity. References have been provided to support the text in the revised manuscript (page 12, lines 266-267).

Comment 12: I understand that the reason why you make more emphasis on miR-128 is that this one seems to have a greater effect than miR-195, and miR-195 has an indirect role in the regulation of NPRA. However, I consider the findings about miR-195 part of your, and it should be included as a conclusion.  

Response: I thank the reviewer for insightful comments. We have incorporated the indirect role of miR-195 in the regulation of NPRA in the revised manuscript. 

MATERIALS AND METHODS.

Comment 13: First part of Material and Methods section (Line 402-415) should be included in a specific subsection and also reorganized.

Response: As suggested by the reviewer, to the first part of Materials and Methods section “Materials” has been added as a subsection in the revised manuscript (page 15, line 323).

Comment 14: In the methodology have been included many references for the same techniques from your laboratory or similar laboratory. Under my concern, the citations should be reduced. For example, Line 452-453 include four references for the same protocol.

Response: In accordance with the reviewer’s comments, we have tried to reduce the multiple citations of the references for the techniques in the Methods section of the revised manuscript.

Comment 15: Check all the sections to avoid typing mistakes. For example, ‘ºC’ (Line 473; 490), ‘CO2’ (Line 506), “MiRNA” (Line 508), etc. Also is described in Line 516 that ‘The nature sequences are present in bold an underline’, however, there is only an underline.

Response: We have edited the manuscripts and have corrected the errors in the text of the revised manuscript. The multiple references in the Materials and Methods section have been reduced and corrected in the amended manuscript.

Comment 16: In Material and Methods should be included all the extra information provided in the Figure legends and removed from that as I mentioned above.

Response: As suggested by the reviewer, the extra information related to methods in the Figure legends have been removed. The Figure legends have been simplified and made succinct in the revised manuscript.

BIBLIOGRAPHY

Comment 17: Please, check all the bibliography and references provided to avoid mistakes. For example, Reference 8 is out of the text, Reference 93 and 94 are the same repeated, etc.

Response: We thank the reviewer, we have corrected the references in the revised manuscript.

Reviewer 2 Report

The work carried out is very interesting and suitable for the biochemical and physiological co-relation of cardiac hormones in their participation and regulation through Guanylyl cyclase in organic homeostasis. Of wich very was known.

Author Response

REVIEWER 2

We thank the reviewer for the valuable time to review our manuscript. We greatly appreciated the positive comments and support for our work.

Reviewer 3 Report

Title: Ligand-Dependent Downregulation of Guanylyl Cyclase/Atrial  Natriuretic Peptide Receptor-A: Role of microRNA-128

Reviewer Comments:

line 19:  "c-ANF (truncated ANF)" do you mean ANP here?

Line 45: "and display the most variability in the primary structure, whereas CNP is highly conserved."

Comment: citation needed

Line 51: suggest change "and guanylyl cyclase" to "and a guanylyl cyclase"

Line 57: Is there a reason why authors include both names of the NP receptors in multiple locations within text?  e.g. guanylyl cyclase/natriuretic peptide receptor-A (GC-A/NPRA).  Should just stick to one name for clarity (e.g. NPRA, NPRB etc)

Line 67: introduction (3rd paragraph):  downregulation of NPRA is outlined towards end of introduction, why would this be of general interest to the reader? why do this experiment?

Line 76: " several growth factors [52–55], certain neurotransmitters [56], and physiological and pathophysiological milieu [57–59]."

Comment: this is vague, if you want to mention this you should specify what growth factors/neurotransmitters etc.

Line 78: "Physiological changes in intact animals are known to raise circulating levels of ANP"

Comment: Physiological changes is also vague and could mean many things.  should give a few examples.

Line 88: introduction (4rd paragraph):  miRs seems to come out of nowhere, should include why authors think miRs regulate Natriuretic Peptide function (e.g. there are many studies showing this link PMID: 23867623; PMID: 29698509; PMID: 27185878 etc)

Line 93: "Specifically, miR-128 has been shown to have an important role in cardiovascular function and development during embryogenesis"

Comment: this isn't very specific, please outline very briefly example(s).

Line 105: suggest change "Pretreatment with 100 nM ANP showed a time- and dose-dependent reduction in 125I-ANP binding to NPRA in HEK-293 cells (90%) after 24 h of exposure (Figure 1A)."  to "Up to 24 h of pretreatment with 100 nM ANP caused a time-dependent reduction in 125I-ANP binding to NPRA in HEK-293 cells (90%) (Figure 1A)."

Line 108: "Both HEK-293 and MA-10 cells contained predominantly NPRA."  You don't measure this here? should be moved to discussion

Line 110: "We visualized the internalization of eGFP-tagged receptor after cells were treated with 100 nM ANP for 5, 10, and 30 min"  Random single images aren't compelling, can this be quantified?

Line 125: "Cells treated with 100 nM ANP showed the appearance of endocytic vesicles after 5 min; these vesicles gradually decreased over 30 min, which is a visual characteristic of internalized membrane receptors within the cytoplasm. In contrast, endocytic vesicles were absent in untreated control cells." 

These observations should be better highlighted in images, can these changes be quantified?

Line 128: suggest change "Pretreatment with 100 nM BNP also reduced receptor binding" to "Pretreatment with 100 nM BNP also reduced 125I-ANP binding to HEK-293 and MA-10 cells"

Line 130: suggest change "with either CNP or c-ANF" to "with 100nM of either CNP or c-ANF"

Line 137: "1X10-7 M" please use same concentration units as for ANP (i.e. 100 nM)

Line 143: suggest change "GC activity of the receptor" to "GC activity of its receptor"

Line 144: suggest change "we examined the effect of ANP pretreatment in HEK-293 and MA-10 cells." to "we quantified GC activity following ANP pretreatment of HEK-293 and MA-10 cells (Figure 3)."

Line 147: suggest change "blocked the downregulation of NPRA" to " blocked the downregulation of NPRA as assessed by 125I-ANP binding"

Line 160: general comment.  Can effects of A71915 and KT5823 be displayed on a single bar graph? so comparisons between the two drugs can be made? (i.e. combine Fig 4A and Fig4 B)

Line 162: " A) Confluent HEK-293 cells and B) MA-10 cells “ Incorrect legend, A and B contain both cell types?

Line 163: "pretreated with varying concentrations of ANP" figures don't indicate any variation in ANP concentration please list concentration used in figure legend (or label figure directly)

Line 169: Comment: lines in graphs A and B are not labelled as having any significant differences. typo?

Line 193: "Okadaic acid (OA) inhibited the protein phosphorylation that also blocked the binding of ANP to NPRA (Figure 6C). " phosphorylation was not measured.  Need to state here more clearly that Okadaic acid blocked the enhancement of 125I-ANP binding to MA-10 cells following CPT-cGMP pretreatment.

Line 199: "A) Confluent MA-10 cells were pretreated with varying concentrations of 8-(4-chlorophenylthio)-cGMP (CPT-cGMP) for the indicated times at 37oC." 

Comment: figure 6A doesn't indicate any variation in CPT-cGMP concentration please list concentration used in figure legend (or label figure directly)

Line 207: "Confluent MA-10 cells were pretreated with CPT-cGMP in the presence or absence of indicated concentrations of okadaic acid (OA) for the indicated times at 37oC. "

Comment: what concentration of CPT-cGMP was used? no times were indicated in Figure 6C?

Line 214: "In the present study, we selected microRNAs since they play a significant role in the biological action of GC-A/NPRA. Specifically, we selected miR-128 and miR-195, which have been shown to regulate the ANP target cell functions."

Comment: This should be in introduction text

Line 216: " Using miRNA target predictor programs (Target Scan release 7.1) [69], we searched for miRs with a predicted target pairing sequence on untranslated region (UTR) of the Npr1 gene."

Comment: How the miRs were selected should form a section in the methods.   

Line 218: "of the Npr1 gene"

Comment: was Npr1 gene (mouse) sequence used instead of human (NPR1) sequence?

Line 225: suggest change "After short-term treatment with ANP, " to "After short-term (0-30 min) treatment with ANP, "

Line 257: suggest change "cells with prolonged exposure to ANP" to "cells with prolonged (≤ 24 hours) exposure to ANP"

Line 241: Figure 8 comment: Could you combine the densitometry analysis graphs (i.e. Fig8 B, D, F; and Fig8 H, J, L) to allow comparisons between the different treatments?

Line 244: suggest change "After short-term treatments" to "After short-term (0-30 min) treatments"

Line 249: suggest change "After long-term treatments" to "After long-term (0-24 hr) treatments"

Line 331: this paragraph is lacking in citations.

Line 354: "Our present results show that miR-128 significantly repressed NPRA protein levels."

Comment: Not sure I agree considering how marked the reduction is following the control miR.  Please discuss this.

Line 367: Comment: Don't feel that figure 9 is needed as it doesn't offer any significant elucidation to what is already covered in text.

Line 412:  Cells were cultured in 95% O2?

Line 476: suggest change "described earlier" to ""described previously"

Line 476: "eGFP-NPRA" where was this sourced? How were cells generated containing this?

Line 477: "grown for 2 days" using what media?

Line 483: how was florescence intensity of cGMP quantified to generate Figure 5?  Please outline here. 

Line 490: "anti-cGMP antibodies" where were these sourced?

Line 491: "anti-rabbit IgG (1:500) conjugated with DyLightTM405" where were these sourced?

Line 499: "(sequential scans with wavelengths set up as green, 488-510; blue, 400-421)".  Information not clear, what were excitation and emission wavelengths used?

Line 502: "The pinhole was adjusted to keep the same size of z-optical sections (1-μm z-axis) for both channels."  This suggests z-stacks were collected, how many z-sections were collected, how were images processed?

Line 542: "GraphPad Prism software" what version was used?

References

100 seems a bit much for a research article, with many of them quite old for what they are being used to cite (e.g. ref 3 is >40 y/o).  please condense reference count down, focusing on more recent articles especially pointing readers to relevant recent reviews.

Author Response

REVIEWER 3

We thank the reviewer for the valuable time to review our manuscript. We greatly appreciated the positive and insightful comments, which have helped to improve the contents of this manuscript to a great extent.

Comment 1: Line 19:"c-ANF (truncated ANF)" do you mean ANP here?

Response: It is written as “c-ANF”, because the original nomenclature is still used to identify the truncated molecule “c-ANF” than full length peptide (page 2, line 34).

Comment 2: Line 45: "and display the most variability in the primary structure, whereas CNP is highly conserved."

Response: As suggested by the reviewer, the reference has been cited in the revised manuscript (page 3, lines 59).

Comment 3: Line 51: suggest change "and guanylyl cyclase" to "and a guanylyl cyclase"

Response: In accordance with the reviewer’s comments, the correction has been incorporated in the revised manuscript (page 3, lines 64).

Comment 4: Line 57: Is there a reason why authors include both names of the NP receptors in multiple locations within text?  e.g. guanylyl cyclase/natriuretic peptide receptor-A (GC-A/NPRA).  Should just stick to one name for clarity (e.g. NPRA, NPRB etc)

Response: As suggested by the reviewer and to keep uniformity, NPRA and NPRB have been used through the text in the revised manuscript.

Comment 5: Line 67: introduction (3rd paragraph):  downregulation of NPRA is outlined towards end of introduction, why would this be of general interest to the reader? why do this experiment?

Response: As indicated by the reviewer, we have introduced the phrase “downregulation” at the appropriate place in the “Introduction” section (2nd paragraph) of the revised manuscript (page 4, lines 72-74).  

Comment 6: Line 76: " several growth factors [52–55], certain neurotransmitters [56], and physiological and pathophysiological milieu [57–59]."

Response: In response to the reviewers comment, the specific growth factors and neurotransmitters have been included in the revised manuscript.

Comment 7: Line 78: "Physiological changes in intact animals are known to raise circulating levels of ANP"

Response:The physiological responses” has been extended in the revised manuscript (page 4, lines 79-85).

Comment 8: Line 88: introduction (4rd paragraph):  miRs seems to come out of nowhere, should include why authors think miRs regulate Natriuretic Peptide function (e.g. there are many studies showing this link PMID: 23867623; PMID: 29698509; PMID: 27185878 etc)

Response: We thank the reviewer for the comment and locating the new references, which we have cited in the revised manuscript to make a smooth transition of the description of microRNA in the “Introduction” section of the text in the revised manuscript (page 4, lines 86-87).

Comment 9: Line 93: "Specifically, miR-128 has been shown to have an important role in cardiovascular function and development during embryogenesis"

Response: As indicated by the reviewer, the specific function of mir-128 has been included in the revised manuscript (page 4, lines 91-92; page 5, lines 93-95).

Comment 10: Line 105: suggest change "Pretreatment with 100 nM ANP showed a time- and dose-dependent reduction in 125I-ANP binding to NPRA in HEK-293 cells (90%) after 24 h of exposure (Figure 1A)."  to "Up to 24 h of pretreatment with 100 nM ANP caused a time-dependent reduction in 125I-ANP binding to NPRA in HEK-293 cells (90%) (Figure 1A)."

Response: We thank the reviewer for the helpful comments. The suggested changes have been incorporated in the revised manuscript (page 5, lines 108-111).

Comment 11: Line 108: "Both HEK-293 and MA-10 cells contained predominantly NPRA."  You don't measure this here? should be moved to discussion

Response: As suggested by the reviewer the phrase “Both HEK-293 and MA-10 cells contained predominantly NPRA” has been removed from the “Results” section to the “Discussion” section of the revised manuscript (page 9, lines 198-200).

Comment 12: Line 110: "We visualized the internalization of eGFP-tagged receptor after cells were treated with 100 nM ANP for 5, 10, and 30 min" Random single images aren't compelling, can this be quantified?

Response: At least 100 untreated or treated cells (three coverslips per experiment/condition) were scored, using a confocal microscope (Leica) with a 63×/1.4 NA Plan-Apochromat oil immersion objective lens. As recommended by the reviewer, we have quantified the fluorescence intensity and new results have been presented in Figure 1, panel D in the revised manuscript (page 43).

Comment 13: Line 125: "Cells treated with 100 nM ANP showed the appearance of endocytic vesicles after 5 min; these vesicles gradually decreased over 30 min, which is a visual characteristic of internalized membrane receptors within the cytoplasm. In contrast, endocytic vesicles were absent in untreated control cells." 

These observations should be better highlighted in images, can these changes be quantified?

Response: Endocytic vesicles were observed after 5, 10, and 30 min of treatment. The magnified images of areas have been indicated in red squares (Figure 1C) and quantitative fluorescence intensity is presented in Figure 1D in the revised manuscript (page 43).

Comment 14: Line 128: suggest change "Pretreatment with 100 nM BNP also reduced receptor binding" to "Pretreatment with 100 nM BNP also reduced 125I-ANP binding to HEK-293 and MA-10 cells"

Response: The suggested change by the reviewer “pretreatment with 100 nM BNP also reduced 125I-ANP binding to HEK and MA-10 cells” has been incorporated in the revised manuscript. (page 6, lines 116-117).

Comment 15: Line 130: suggest change "with either CNP or c-ANF" to "with 100nM of either CNP or c-ANF"

Response: The suggested change by the reviewer “with 100 nM of either CNP or c-ANF” has been incorporated in the revised manuscript (page 6, lines 117-118).

Comment 16: Line 137: "1X10-7 M" please use same concentration units as for ANP (i.e. 100 nM)

Response: As suggested by the reviewer, the concentration unit 1X10-7 M has been changed to 100 nM in legend to Figure 2 (page 38, line 844).

Comment 17: Line 143: suggest change "GC activity of the receptor" to "GC activity of its receptor"

Response: The suggested change “GC activity of its receptor” has been incorporated in the revised manuscript (page 6, line 130).

Comment 18: Line 144: suggest change "we examined the effect of ANP pretreatment in HEK-293 and MA-10 cells." to "we quantified GC activity following ANP pretreatment of HEK-293 and MA-10 cells (Figure 3)."

Response: The suggested change “we quantified GC activity following ANP pretreatment of HEK-293 and MA-10 cells (Figure 3)” has been incorporated in the revised manuscript (page 6, line 130-131).

Comment 19: Line 147: suggest change "blocked the downregulation of NPRA" to" blocked the downregulation of NPRA as assessed by 125I-ANP binding"

Response: The suggested change “blocked the down regulation of NPRA as assessed by 125I-ANP binding” has been incorporated in the revised manuscript (page 6, line 134).

Comment 20: Line 160: general comment.  Can effects of A71915 and KT5823 be displayed on a single bar graph? so comparisons between the two drugs can be made? (i.e. combine Fig 4A and Fig4 B)

Response: We appreciate the reviewer comment. Combining Figure 4A and 4B on a single bar graph would increase number of bar lines in the graph in one Figure, and compromise the clarity. Thus, we have kept both Figure 4A and 4B panels in the manuscript.

Comment 21: Line 162: " A) Confluent HEK-293 cells and B) MA-10 cells “ Incorrect legend, A and B contain both cell types?

Response: We thank the reviewer for the insightful comment. Now the Figure 4, A) indicates the treatment of NPRA antagonist, A71915 and B) indicates the treatment of PKG inhibitor, KT-5823 in the legend to Figure 4 of the amended manuscript (page 6, lines 133-138).

Comment 22: Line 163: "pretreated with varying concentrations of ANP" figures don't indicate any variation in ANP concentration please list concentration used in figure legend (or label figure directly)

Response: We thank the reviewer for the comment. We apologize for the inadvertent error, now we have incorporated the concentration of ANP (100 nm) in the legend to Figure 4 in the revised manuscript (page 39, line 856).

Comment 23: Line 169: Comment: lines in graphs A and B are not labelled as having any significant differences. typo?

Response: We thank the reviewer for the insightful comment. We have added the significance symbols in figure 4 A and B in the revised manuscript. (Figure 4A and B).

Comment 24: Line 193: "Okadaic acid (OA) inhibited the protein phosphorylation that also blocked the binding of ANP to NPRA (Figure 6C). " phosphorylation was not measured.  Need to state here more clearly that Okadaic acid blocked the enhancement of 125I-ANP binding to MA-10 cells following CPT-cGMP pretreatment.

Response: As suggested by the reviewer, we have corrected the sentence to indicate “Okadaic acid blocked the enhancement of 125I-ANP binding in MA cells following CPT-cGMP treatment in the revised manuscript (page 7, lines 159-160).

Comment 25: Line 199: "A) Confluent MA-10 cells were pretreated with varying concentrations of 8-(4-chlorophenylthio)-cGMP (CPT-cGMP) for the indicated times at 37oC." 

Comment: figure 6A doesn't indicate any variation in CPT-cGMP concentration please list concentration used in figure legend (or label figure directly)

Response: As indicated by the reviewer, the concentration of CPT-cGMP (1µm) has been incorporated in the revised manuscript (page 7, line 160).

Comment 26: Line 207: "Confluent MA-10 cells were pretreated with CPT-cGMP in the presence or absence of indicated concentrations of okadaic acid (OA) for the indicated times at 37oC. "

Comment: what concentration of CPT-cGMP was used? no times were indicated in Figure 6C?

Response: The cells were treated with 1µm CPT-cGMP for 24h at 37°C. The concentration of CPT-cGMP (1µm) and treatment time (24h) have been incorporated in the revised manuscript (page7, lines 160).

Comment 27: Line 214: "In the present study, we selected microRNAs since they play a significant role in the biological action of GC-A/NPRA. Specifically, we selected miR-128 and miR-195, which have been shown to regulate the ANP target cell functions."

Comment: This should be in introduction text

Response: As suggested by the reviewer, in the present study the phrase “we selected microRNAs since they play a significant role in the biological action of GC-A/NPRA” has been moved from the “Results” section to the “Introduction” of the revised manuscript (page 5, lines 97-100).

Comment 28: Line 216: " Using miRNA target predictor programs (Target Scan release 7.1) [69], we searched for miRs with a predicted target pairing sequence on untranslated region (UTR) of the Npr1 gene."

Comment: How the miRs were selected should form a section in the methods.   

Response: As suggested by the reviewer, the miRNA target predictor program has been moved to the “Methods” section in the revised manuscript (page 20, lines 437-459; page 21, lines 460-464).

Comment 29: Line 218: "of the Npr1 gene"

Comment: was Npr1 gene (mouse) sequence used instead of human (NPR1) sequence?

Response: In the current studies, instead the human (Npr1) sequence we used mouse Npr1 sequence (Garg et al., 2002).

Comment 30: Line 225: suggest change"After short-term treatment with ANP, " to "After short-term (0-30 min) treatment with ANP, "

Response: As suggested by the reviewer the phrase after short-term treatment with ANP” has been changed to “after short-term (0-30min) treatment with ANP” in the revised manuscript (page 8, line 168).

Comment 31: Line 257: suggest change"cells with prolonged exposure to ANP" to "cells with prolonged (≤ 24 hours) exposure to ANP"

Response: As suggested by the reviewer, the phrase “cells with prolonged exposure to ANP” has been changed to “cells with prolonged (≤ 24 hours) exposure to ANP” in the revised manuscript (legend to Figure 7).

Comment 32: Line 241: Figure 8 comment: Could you combine the densitometry analysis graphs (i.e. Fig8 B, D, F; and Fig8 H, J, L) to allow comparisons between the different treatments?

Response: We thank the reviewer for the comment. However, we feel that the direct correlative comparisons of gel bands with densitometry data would be difficult to compare with all densitometry analysis combined in one graph. Thus, we have kept as original labels in the Figure 8 (revised Figure 7) in the amended manuscript.

Comment 33: Line 244: suggest change "After short-term treatments" to "After short-term (0-30 min) treatments"

Response: As suggested by the reviewer the phrase “after short-term treatment” has been changed to “after short-term (0-30 min)”. Please also see our response to comment # 25. The correction has been incorporated in the revised manuscript (legend to Figure 7).

Comment 34: Line 249: suggest change "After long-term treatments" to "After long-term (0-24 hr) treatments"

Response: As suggested by the reviewer, the phrase “after long-term treatment” has been changed to “after long-term (0-24 h) treatment”. The correction has been incorporated in the revised manuscript.

Comment 35: Line 331: this paragraph is lacking in citations.

Response: We have corrected the statements reflecting the discussion on the current results, thus we did not use the citation of the references by others, however, in continuation, we have discussed previous findings in relation to our present studies.

Comment 36: Line 354: "Our present results show that miR-128 significantly repressed NPRA protein levels."

Comment: Not sure, I agree considering how marked the reduction is following the control miR.  Please discuss this.

Response: We agree with the reviewer’s comment that the decrease is only modest; however we feel that these observed decreases are dynamic in nature and there is not a complete inhibition by miRNAs. Since there seem to be continuous synthesis and degradation of NPRA mRNA and protein products, therefore, at this point, we do not have enough tools to capture actual protein levels. In future, we would like to apply more advanced and alternative approaches to resolve these difficulties.

Comment 37: Line 367: Comment: Don't feel that figure 9 is needed as it doesn't offer any significant elucidation to what is already covered in text.

Response: In accordance with the reviewer’s comments, we have deleted the original Figure 9 in the revised manuscript.

Comment 38: Line 412:  Cells were cultured in 95% O2?

Response: We have corrected 95% O2to 95% air.

Comment 39: Line 476: suggest change "described earlier" to ""described previously"

Response: As indicated by the reviewer, the phrase “described earlier” has been changed to “previously described.”

Comment 40: Line 476: "eGFP-NPRA" where was this sourced? How were cells generated containing this?

Response: The eGFP-NPRA construct was prepared and validated in HEK-293 cells in our laboratory (Mani et al., 2015; Mani et al., 2016). The references have been cited in the revised manuscript (page 18, lines 399-400).

Comment 41: Line 477: "grown for 2 days" using what media?

Response: Cells were grown for 2 days in DMEM medium containing 10% calf serum. The correction has been incorporated in the revised manuscript (page 18, line 401).

Comment 42: Line 483: how was florescence intensity of cGMP quantified to generate Figure 5?  Please outline here. 

Response: The cGMP immunofluorescence localization was done using confocal microscopy. A brief statement has been incorporated in the revised manuscript (page 19, lines 419-420).

Comment 43: Line 490: "anti-cGMP antibodies" where were these sourced?

Response: Anti-cGMP antibodies were received from the Antibodies Online, Inc., Atlanta, GA, USA, which has been incorporated in the revised manuscript (page 19, lines 414-415).

Comment 44: Line 491: "anti-rabbit IgG (1:500) conjugated with DyLightTM405" where were these sourced?

Response: Goat-anti-rabbit IgG (1:500) conjugated with Dylight TM405 were obtained from Jackson Immuno Research Laboratories, Inc. (West Grove, PA, USA). The information has been incorporated in the revised manuscript (page 15, line 332-333)

Comment 45: Line 499: "(sequential scans with wavelengths set up as green, 488-510; blue, 400-421)".  Information not clear, what were excitation and emission wavelengths used?

Response: For the green channel (eGFP-tagged NPRA) excitation was 488 nm and emission was 510 nm. Whereas for the blue channel (DyLightTM405anti-rabit antibody for cGMP immunofluorescence), excitation was 400 nm and emission was 421 nm. The information have been appropriately incorporated in the revised manuscript (page 19, lines 428-430).

Comment 46: Line 502: "The pinhole was adjusted to keep the same size of z-optical sections (1-μm z-axis) for both channels."  This suggests z-stacks were collected, how many z-sections were collected, how were images processed?

Response: In all experiments, images of cells were acquired as single mid-cellular optical sections and averaged over eight scans/frame. MetaMorph software (Molecular Devices) was used for to image process.

Comment 47: Line 542: "GraphPad Prism software" what version was used?

Response: We have used GraphPad Prism software with version 6.0, which has been included in the revised manuscript (page 22, line 484).

References

Comment 48: 100 seems a bit much for a research article, with many of them quite old for what they are being used to cite (e.g. ref 3 is >40 y/o).  Please condense reference count down, focusing on more recent articles especially pointing readers to relevant recent reviews.

Response: We have tried to evaluate the reference citations in the current manuscript. Some of the older references have been deleted in the revised manuscript. The original reference 3 is an original paper with the discovery of ANF/AMP by de Bold and his colleagues.  Since, the reference by de Bold et al., 1985 is cited; the reference #3 has been deleted in the revised manuscript.

Round 2

Reviewer 1 Report

The authors have satisfactorily addressed all my revisions. The manuscript ‘Ligand-dependent downregulation of guanylyl cyclase/atrial natriuretic peptide receptor-A: role of microRNA-128’ demonstrated an interesting new insight into the cardiac natriuretic peptide hormone family and its role and provide new information about the regulation of miRNA in this topic. Thus, I consider that the manuscript provides useful scientific information and keeps the quality standards of the IJMS journal, and consequently, it is ready for being accepted. 

Congratulations on your work.  

Author Response

We thank the reviewer for the valuable time to review our manuscript. We greatly appreciated the positive comments and support for our work.